# Investigation of the Physical Mechanism of Acoustic Attenuation in Viscous Isotropic Solids

**DOI:** 10.3390/mi13091526

**Published:** 2022-09-15

**Authors:** Lin Fa, Lili Li, Hong Gong, Wenhui Chen, Jing Jiang, Guoqiang You, Jifeng Liang, Yandong Zhang, Meishan Zhao

**Affiliations:** 1School of Electronic Engineering, Xi’an University of Posts and Telecommunications, Xi’an 710121, China; 2School of Information Engineering, Xi’an Fanyi University, Xi’an 710105, China; 3Logging Institute, CNPC Logging, Xi’an 710121, China; 4James Franck Institute and Department of Chemistry, The University of Chicago, Chicago, IL 60637, USA

**Keywords:** damping attenuation, propagation attenuation, particle displacement, acoustic attenuation characteristics, dispersion characteristics

## Abstract

The traditional acoustic attenuation coefficient is derived from an analogy of the attenuation of an electromagnetic wave propagating inside a non-ideal medium, featuring only the attenuation of wave propagation. Nonetheless, the particles inside viscous solids have mass, vibrating energy, viscosity, and inertia of motion, and they go through transient and damping attenuation processes. Based on the long-wavelength approximation, in this paper, we use the energy conservation law to analyze the effect of the viscosity of the medium on acoustic attenuation. We derive the acoustic attenuation coefficient by combinations of the dynamical equation of a solid in an acoustic field with conventional longitudinal wave propagation under a spring oscillator model. Considering the attenuation of propagating waves and the damping attenuation of particle vibration, we develop a frequency dispersion relation of phase velocity for the longitudinal wave propagating inside viscous solid media. We find that the acoustic impulse response and vibrational system function depends on the physical properties of the viscous solid media and their internal structure. Combined with system function, the impulse response can be an excellent tool to invert the physical properties of solids and their internal structures. We select a well-known rock sample for analysis, calculate the impulse response and vibrational system function, and reveal new physical insight into creating acoustic attenuation and frequency dispersion of phase velocity. The results showed that the newly developed acoustic attenuation coefficients enjoy a substantial improvement over the conventional acoustic attenuation coefficients reported in the literature, which is essential for industrial applications; so are the dispersion characteristics.

## 1. Introduction

Acoustic attenuation created by acoustic waves propagating in a medium is one of the most natural fundamental physical phenomena in acoustics. A vibrating particle in the medium acts on its neighbor particle through internal stress and causes an adjacent particle to move. This process is ongoing repeatedly, leading to the propagation of fluctuations and acoustic waves inside the solid. In nature, all media, either fluids or solids, have a certain degree of viscosity and thermal conductivity, which can cause acoustic attenuation. On the other hand, medium compressibility means that the medium is composed of many particles, and acoustic scattering from particles can also cause acoustic attenuation. Researchers have conducted extensive theoretical, experimental, and applied research to grasp the physical mechanism of creating acoustic attenuation from microscopic and macroscopic perspectives.

Atkinson et al. studied the multiparticle interactions in dense suspensions, acoustic wave speed, and attenuation in suspensions [1], also reported by Peter et al. [2]. Gibson and Toksöz proposed and analyzed a model for attenuation of acoustic waves in suspensions that includes an energy loss due to viscous fluid flow around spherical particles [3]. Babick and Richter analyzed the effect of visco-inertial coupling on sound attenuation [4], and Verma et al. [5] reported the effect of thermal conductivity. Their results showed that acoustic attenuation mainly came from the contribution of viscosity for smaller particles compared to the wavelength. The size of pores in solid media (rocks or sediments) is usually much smaller than the wavelength of either the general acoustic signal or seismic exploration signal. Therefore, during the studies of acoustic attenuation of an acoustic wave propagating in either rocks or sediments as an analog to suspensions, many scientists neglected the effects of the thermal conductivity and scattering of the particles on attenuation, considering only the contribution of viscosity to acoustic attenuation.

Geophysicists have reported studies on acoustic attenuation in various porous media. With respect to the field of onshore and marine geology, Yang et al. measured the acoustic attenuation coefficient of sediments in the frequency range of 2–8 kHz [6]; Hefner et al. studied the effect of porosity on acoustic attenuation in sandstone sediments at high frequency [7]; Zheng et al. studied the acoustic attenuation of gaseous deposits [8]; so are Wang et al. who conducted an experimental study on acoustic velocity and attenuation of gaseous sediments [9]; Tang et al. performed experiment in studying the relationship between formation fracture width and acoustic attenuation coefficient [10]; Meyer et al. studied the acoustic attenuation within the glacier with a frequency range from 2 to 35 kHz [11]; Cooper studied the energy loss and attenuation of seismic waves propagating in viscous solids [12]; Long performed an experimental study on acoustic attenuation of seafloor sediments [13]; Jiang et al. measured and studied low-frequency acoustic attenuation in marine sediments [14]; Zou et al. proposed a method for calculating the attenuation of the first wave of acoustic signals propagating in seabed sediments [15]; Zimmer et al. experimentally measured the relationships of both acoustic velocity and acoustic attenuation coefficient of seabed sand-gravel versus the frequency within the frequency range of 1 kHz to 400 kHz [16]; Wan et al. measured the attenuation of long-distance wideband acoustic signals in the undersea situation [17]. However, as reported by Atkinson et al., acoustic wave propagation in porous and suspension media, while similar, has the following significant differences: porous media exhibit elastic resistance to shear stresses, but suspension media typically do not, while both can sustain isotropic stresses [1].

Similar to an electromagnetic wave propagating inside a non-ideal medium, an acoustic wave propagates inside a viscous medium. These two propagations have some similarities but also some differences. As is known, an electromagnetic wave has only energy but does not have mass, and we use the Maxwell equation to solve the problems in an electromagnetic field. An electromagnetic wave propagating in a non-ideal (non-zero electrical conductivity) medium has propagation attenuation only, without damping attenuation, i.e., its attenuation is only about the propagation distance and is independent of time. The vibrational particle corresponding to the acoustic wave inside a solid medium is a substance with energy and mass, containing the damping attenuation of particle vibration and propagation attenuation of fluctuation and solving the problems of Newton’s law. In other words, at any position in space, the generation or disappearance of a harmonic electromagnetic field does not experience a transient process of electromagnetic wave (electromagnetic field intensity) changing with time. At the same time, the harmonic-sinusoidal vibration of particles follows Newton’s inertia theorem. There is a transient process as follows: the particle transits from a static state to a steady harmonic-sinusoidal vibrational state and vice versa.

Scientists have reported studies of damped oscillations of particles. For example, Maity et al. found that damped oscillation can enhance the system time interval of acoustic imaging sensors [18]. Fa and Zhao et al. studied the damping attenuation properties of several types of piezoelectric transducers [19,20,21] but have not yet combined damping attenuation of particle vibration with propagation attenuation of fluctuation for studies of acoustic attenuation.

Most of the published work in the literature used a simple analogy of the attenuation generated by an electromagnetic wave propagating in non-ideal media to study the attenuation of the acoustic wave without enough understanding of the difference in attenuation caused by the different characteristics of the acoustic wave and electromagnetic wave [22].

This paper reports the newly derived acoustic attenuation coefficient by combining the dynamical equation of solids in an acoustic field with conventional longitudinal wave propagation under a spring oscillator model and a frequency dispersion relation of phase velocity for the longitudinal wave propagating inside viscous solid media. We selected a well-known rock sample for analysis. We calculated the system impulse response and vibrational function, revealing new physical insight for creating propagation attenuation and frequency dispersion of phase velocity. We found that the acoustic impulse response and the system function of vibration particles depended on the physical properties of the viscous solid media and their internal structure. The calculated results showed a significant improvement over the conventional acoustic attenuation coefficient reported in the literature [22], which is essential for industrial applications; so are the dispersion characteristics.

## 2. Physics Model

A perfectly elastic solid is just a hypothetical ideal case. There is no naturally existing perfect medium with zero conductivity, and there is no ideal solid medium with zero viscosity, i.e., all solid media have some degree of stickiness. An electromagnetic wave has energy but does not have mass. When an electromagnetic wave propagates inside a vacuum, there is no internal energy loss. Such a wave propagates with equal amplitude without frequency dispersion. When a harmonic electromagnetic wave propagates in the medium, there is no damping attenuation. Still, the thermal loss caused by the non-zero conductivity of the medium will cause the electromagnetic wave to create propagation attenuation.

The “vibration particle” in acoustics is a tangible substance with mass and energy. Unlike electromagnetic waves, acoustic waves cannot propagate in a vacuum but only in a medium. Every solid medium has a specific viscosity, which leads to frictional force and heat dissipation whenever the particle inside a viscous medium starts to vibrate. Frictional force always causes particle vibration amplitude to decrease with time and the wave amplitude to decrease with the increase in propagation distance. Therefore, there is a certain degree of internal energy loss in the particle vibration process near its equilibrium position and fluctuation propagation.

The propagation attenuation and particle vibrational damping attenuation inside viscous solids significantly influence acoustic wave propagation. A stationary particle begins to vibrate under the action of an external harmonic force. Due to the particle’s inertia and the friction force’s action, there is a transient process in the transition from a stationary particle to a steady harmonic vibration. The frequency spectrum corresponding to this process is the intrinsic noise produced by vibration particles inside viscous solids. Therefore, the particle vibration inside dense media contains the frequency component of the external harmonic force and the frequency component generated by the corresponding particle vibration transient process.

Damping attenuation is a measure that describes the vibration amplitude attenuation of the particle near its equilibrium position in the time domain. In contrast, propagation attenuation is a measure that describes the amplitude attenuation of a fluctuation propagating in space with increasing propagation distance.

Based on the elastic constitutive relation of viscous solid [22], we established the theoretical relationship between the propagation attenuation of fluctuation and the damping attenuation of particle vibration, studied the influence of viscosity on the damping attenuation of particle vibration and the propagation attenuation, and gave a new analytic expression of the acoustic attenuation coefficient and a theoretical explanation of the physical mechanism of creating acoustic attenuation.

### 2.1. Damping Attenuation of Particle Vibration in Viscous Isotropic Solid Media

Let us investigate acoustic attenuation by first studying particle vibration’s damping attenuation.

When a body moves in a viscous fluid, it is subject to frictional force. The magnitude of this frictional resistance is related to the viscosity of the medium and the body’s shape, size, and movement speed. For example, the frictional resistance force of a small ball moving at a constant speed in a viscous liquid is as follows [23]:(1)$$f=6\pi r\eta v=R_{0}v$$

There are several important parameters: *r* is the radius and *v* is the moving speed of the small ball; *η* is the viscosity coefficient of the liquid; $R_{0}=6\pi r\eta$ is the frictional resistance.

Particles in a viscous solid usually satisfy the condition of “long-wavelength approximation,” simplifying the physical analysis of the vibrational particles regardless of the shape of the acoustic elements [24]. Analog to a small moving ball in a viscous fluid, we assume that the vibration particles in viscous media act very similar to a tiny ball. The frictional force experienced by a vibrating particle is proportional to its vibration velocity and the viscous coefficient of the solid medium. The direction of the frictional force is opposite to the moving direction of the particle.

The elastic damping of particle vibration causes stress to be transmitted very complexly within the viscous solid medium. The physical strain, induced by vibrating particles inside viscous media, acts on neighboring particles, causing them to vibrate. Again, this process repeats, leading to the propagation of fluctuation, that is, the propagation of acoustic waves.

Below, we shall discuss only the propagation of a longitudinal wave, which corresponds to the compressive strain of a solid medium. When a particle vibrates, the size and density of the tiny volume element corresponding to the particle will change. From the damped spring oscillator model (see Figure 1a), we can obtain the motion equation of particle vibration as follows:(2)$$m\frac{d^{2}u}{dt^{2}}+R_{m}\frac{du}{dt}+k_{c}u=f_{a}$$

The mass of the particle (*m*) analogies as an inductance in electricity. *k_c_* is the stubbornness coefficient of the spring oscillator; $C_{m}(=1/k_{c})$ corresponds to capacitance; *R_m_* is the frictional resistance, corresponding to resistance; $f_{a}(t)$ is the force acting on the particle, corresponding to a voltage source; *u* is particle displacement, which corresponds to charge; v(=du/dt) corresponds to current.

By adopting a similar approach reported in references [19,20,21] and using the residue theorem to solve the motion equation of the particle in a viscous solid, we can obtain the acoustic impulse response and system function of the particle vibration system as follows:(3)$$h(t)=Ae^{-\beta t}\cos(\omega_{d}t+\theta )\varepsilon (t)$$
(4)$$H(\omega )=H(s){|}_{s=i\omega }=\frac{i\omega C_{m}}{-mC_{m}\omega^{2}+iR_{m}C_{m}\omega +1}$$

In these equations, ω is angular frequency; ε(t) is a unit step function; $\beta =R_{m}/2m$, which is the damping attenuation coefficient of particle vibration; $\omega_{d}=\sqrt{4mC_{m}-{\left(R_{m}C_{m}\right)}^{2}}/2mC_{m}$, which is the frequency of the damping vibration and real (not complex); $A=2C_{m}\sqrt{\beta^{2}+{\left(\omega_{d}\right)}^{2}}$; $\theta =\tan^{-1}\left(-\omega_{d}/\beta \right)$.

The parameter β describes the damping attenuation state of particle vibration. Equations (3) and (4) show that the impulse response and vibrational system function depend on the physical parameters of the viscous solids. Therefore, from Equations (3) and (4), we can use the measured acoustic signal to invert the inherent physical characteristics of viscous solid media and judge the formation’s internal structure.

### 2.2. Damped Elastic Constitutive Relation of Viscous Solid

We use strain and stiffness coefficients of a solid medium to simulate the particle displacement and apply the spring’s stubborn coefficient in the above “physical model of spring oscillators.” We also use the time derivative of the strain and viscosity coefficient matrix to analogy the moving speed of the tiny volume element and the friction resistance, so the stress on the particle inside viscous media can be written by the following [22]:(5)$$T=c:S+\eta :\frac{\partial S}{\partial t}$$
In Equation (5), the double point “:” is the double-dot product operator, indicating the sum of total angular indices; the boldface symbols **T** and **S** are stress tensor and strain tensor, respectively; c and **η** are the stiffness coefficient matrix and viscosity coefficient matrix of the solid, respectively. It is worth noting that Formula (5) considers both the elastic force in the “spring oscillator” model and the frictional force of the vibrating particle caused by the viscosity of the solid.

### 2.3. Propagation Attenuation of Acoustic Waves in Viscous Isotropic Solids

When an acoustic wave propagates in a viscous solid, propagation attenuation of fluctuation and damping attenuation of particle vibration should be considered. Based on Auld’s derivation method for acoustic attenuation coefficient [22], we derived a new expression of the longitudinal wave attenuation coefficient by introducing the factor of particle vibration-damping attenuation in the following.

For an isotropic solid medium, we have its stiffness matrix written as follows:(6)$$c=\left(\left.\begin{array}{cccccc} c_{11} & c_{12} & c_{12} & 0 & 0 & 0\\ c_{12} & c_{11} & c_{12} & 0 & 0 & 0\\ c_{12} & c_{12} & c_{11} & 0 & 0 & 0\\ 0 & 0 & 0 & c_{44} & 0 & 0\\ 0 & 0 & 0 & 0 & c_{44} & 0\\ 0 & 0 & 0 & 0 & 0 & c_{44} \end{array}\right)\right.$$
where *c*_11_ = *c*_12_
*+ 2c*_44_, for *c*_11_, *c*_12_, and *c*_44_, there are only two independent elements.

Analogous to the stiffness matrix of a solid medium, there is also a viscosity coefficient matrix corresponding to the stiffness coefficient matrix, which describes the viscosity of an isotropic solid as follows:(7)$$\eta =\left(\left.\begin{array}{cccccc} \eta_{11} & \eta_{12} & \eta_{12} & 0 & 0 & 0\\ \eta_{12} & \eta_{11} & \eta_{12} & 0 & 0 & 0\\ \eta_{12} & \eta_{12} & \eta_{11} & 0 & 0 & 0\\ 0 & 0 & 0 & \eta_{44} & 0 & 0\\ 0 & 0 & 0 & 0 & \eta_{44} & 0\\ 0 & 0 & 0 & 0 & 0 & \eta_{44} \end{array}\right)\right.$$

A similar relationship exists between viscosity elements ($\eta_{11}=\eta_{12}+2\eta_{44}$), and only two independent variables exist for an isotropic solid’s viscosity coefficient matrix elements. Expanding Formula (5) yields the following equation:(8)$$\left(\begin{array}{c} T_{1}\\ T_{2}\\ T_{3}\\ T_{4}\\ T_{5}\\ T_{6} \end{array}\right)=(\begin{array}{cccccc} c_{11} & c_{12} & c_{12} & 0 & 0 & 0\\ c_{12} & c_{11} & c_{12} & 0 & 0 & 0\\ c_{12} & c_{12} & c_{11} & 0 & 0 & 0\\ 0 & 0 & 0 & c_{44} & 0 & 0\\ 0 & 0 & 0 & 0 & c_{44} & 0\\ 0 & 0 & 0 & 0 & 0 & c_{44} \end{array})\left(\begin{array}{c} S_{1}\\ S_{2}\\ S_{3}\\ S_{4}\\ S_{5}\\ S_{6} \end{array}\right)+(\begin{array}{cccccc} \eta_{11} & \eta_{12} & \eta_{12} & 0 & 0 & 0\\ \eta_{12} & \eta_{11} & \eta_{12} & 0 & 0 & 0\\ \eta_{12} & \eta_{12} & \eta_{11} & 0 & 0 & 0\\ 0 & 0 & 0 & \eta_{44} & 0 & 0\\ 0 & 0 & 0 & 0 & \eta_{44} & 0\\ 0 & 0 & 0 & 0 & 0 & \eta_{44} \end{array})\frac{\partial }{\partial t}\left(\begin{array}{c} S_{1}\\ S_{2}\\ S_{3}\\ S_{4}\\ S_{5}\\ S_{6} \end{array}\right)$$

The abbreviations of strain and stress components subscripts are {1, 2, 3, 4, 5, 6}, and the corresponding full written subscripts are {*xx*, *yy*, *zz*, *yz*, *xz*, *xy*}.

Suppose the longitudinal wave propagates along the *x*-axis direction in the *x-z* plane. The particle vibration corresponding to the longitudinal wave also occurs in the *x-z* plane. Its polarization direction is parallel to the *x*-axis.

An acoustic wave propagating in viscous solids is similar to but different from that of an electromagnetic wave inside a medium with non-zero conductivity. The heat dissipation generated by electromagnetic waves propagating in a solid with non-zero conductivity only comes from propagation attenuation. In contrast, the thermal dissipation, developed in both the particle vibration and the fluctuation propagation processes, comes from the contributions of the damping attenuation of particle vibration and the propagation attenuation of fluctuation. We can assume that the longitudinal wave propagates from the coordinate origin to a space point *x* on the *x*-axis at time *t*, and then its particle displacement is as follows:(9)$$\begin{array}{ll} u & =e_{x}u_{x}(x,t)=e_{x}Ae^{-\beta \left(t-x/v_{p}\right)}e^{-\alpha_{p}x}e^{i\left(\omega t-k_{p}x+\varphi_{p}\right)}H\left(t-x/v_{p}\right)\\ & =\left\{\begin{array}{c} e_{x}Ae^{-\beta t}e^{i\left(\omega t+\varphi_{p}\right))}\;\;\;\;\;\;\;\;\;\;\;\;\;\;\;\;\;\;\;x=0\\ e_{x}Ae^{-\beta \left(t-x/v_{p}\right)}e^{-\alpha_{p}x}e^{i\left(\omega t-k_{p}x+\varphi_{p}\right)}=e_{x}Ae^{\left(i\omega -\beta \right)t}e^{\left(\beta /v_{p}-\alpha_{p}-ik_{p}\right)x}e^{i\varphi_{p}}\;\;x\neq 0,\;t>x/v_{p} \end{array}\right. \end{array}$$
where, $\alpha_{p}$, $k_{p}$ and $v_{p}$ are the acoustic attenuation coefficient, phase coefficient, and phase velocity of a longitudinal wave propagating in a solid medium; *A* is the initial amplitude of particle displacement at *x* = 0 and *t* = 0.

The relationship between strain and particle displacement is [22].
(10)$$S=\nabla_{s}u$$
It yields only one non-zero component of compressive stresses ($S_{1}\neq 0$). All others are zero, including the compressive strains $S_{2}\;{=S}_{3}=0$ and shear strains $S_{4}\;{=S}_{5}\;{=S}_{6}=0$, and we have the following:(11)$$S_{1}{=S}_{xx}=\frac{\partial u_{x}}{\partial x}=\left(\beta /v_{p}-\alpha_{p}-ik_{p}\right)u_{x}(x,t)\;t>x/v_{p}$$

We only need to consider the stress component T_1_ = T*_xx_* in the *x*-direction on the *x*-plane (the plane perpendicular to the *x*-axis). From Formula (8), the lossy elastic constitutive relation between stress component T_1_ and strain component S_1_ can be obtained as follows:(12)$$\begin{array}{l} T_{1}=c_{11}S_{1}+\eta_{11}\frac{\partial S_{1}}{\partial t}=\left[(c_{11}-\eta_{11}\beta )+i\eta_{11}\omega \right]S_{1}\\ \;\;=-\left[(c_{11}-\eta_{11}\beta )+i\eta_{11}\omega \right]\left(\alpha_{p}-\beta /v_{p}+ik_{p}\right)u_{x}(x,t) \end{array}$$

Because the particle in the viscous solid medium is acted on only by the internal stress, which includes both the elastic force of the vibrating particle and the friction force generated by the viscosity of the solid medium, therefore, the dynamical equation can be simplified (from $\nabla \cdot T=\rho \partial^{2}u/\partial t^{2}$) as follows:(13)$$\frac{\partial T_{1}}{\partial x}=\rho \frac{\partial^{2}u_{x}}{\partial t^{2}}$$

The combination of Equation (9) with Equation (11) leads to the following:(14)$$\frac{\partial T_{1}}{\partial x}={\left(\frac{\beta }{v_{p}}-\alpha_{p}-ik_{p}\right)}^{2}\left[(c_{11}-\eta_{11}\beta )+i\eta_{11}\right]u_{x}$$
(15)$$\rho \frac{\partial^{2}u_{x}}{\partial t^{2}}=\rho {\left(i\omega -\beta \right)}^{2}u_{x}$$
Therefore, we have the following:(16)$${\left(\frac{\beta }{v_{p}}-\alpha_{p}-ik_{p}\right)}^{2}\left[(c_{11}-\eta_{11}\beta )+i\eta_{11}\right]=\rho {\left(i\omega -\beta \right)}^{2}$$

Let
(17)$$M={\left(\alpha_{p}-\frac{\beta }{v_{p}}\right)}^{2}-k_{p}{}^{2}$$
(18)$$N=2\left(\alpha_{p}-\frac{\beta }{v_{p}}\right)k_{p}$$
And the real and imaginary parts are equal at both ends of the Equation (15). We can obtain two equations as follows:(19)$$\left(c_{11}-\eta_{11}\beta \right)M-\eta_{11}\omega N=\rho \left(\beta^{2}-\omega^{2}\right)$$
(20)$$\eta_{11}\omega M+\left(c_{11}-\eta_{11}\beta \right)N=-2\rho \beta \omega$$

Solving Equations (19) and (20) results in the following:(21)$$\alpha_{p}=\sqrt{\frac{\sqrt{M^{2}+N^{2}}+M}{2}}+\frac{\beta }{v_{p}}$$
(22)$$k_{p}=\frac{N}{2\sqrt{\frac{\sqrt{M^{2}+N^{2}}+M}{2}}}$$
(23)$$v_{p}=\frac{\omega }{k_{p}}$$
where,
(24)$$Q=\left|\begin{array}{cc} c_{11}-\eta_{11}\beta & -\eta_{11}\omega \\ \eta_{11}\omega & c_{11}-\eta_{11}\beta \end{array}\right|={\left(c_{11}-\eta_{11}\beta \right)}^{2}+{\left(\eta_{11}\omega \right)}^{2}$$
(25)$$M=\frac{\left|\begin{array}{cc} \rho \left(\beta^{2}-\omega^{2}\right) & -\eta_{11}\omega \\ -2\rho \beta \omega & c_{11}-\eta_{11}\beta \end{array}\right|}{Q}=\frac{\rho \left(\beta^{2}-\omega^{2}\right)\left(c_{11}-\eta_{11}\beta \right)-2\rho \beta \eta_{11}\omega^{2}}{{\left(c_{11}-\eta_{11}\beta \right)}^{2}+{\left(\eta_{11}\omega \right)}^{2}}$$
(26)$$N=\frac{\left|\begin{array}{cc} c_{11}-\eta_{11}\beta & \rho \left(\beta^{2}-\omega^{2}\right)\\ \eta_{11}\omega & -2\rho \beta \omega \end{array}\right|}{Q}=-\frac{\rho \eta_{11}\omega \left(\beta^{2}-\omega^{2}\right)+2\rho \beta \omega \left(c_{11}-\eta_{11}\beta \right)}{{\left(c_{11}-\eta_{11}\beta \right)}^{2}+{\left(\eta_{11}\omega \right)}^{2}}$$

The above equations show that the acoustic attenuation coefficient ($\alpha_{p}$) and phase velocity ($v_{p}$) of longitudinal waves are related to their frequency and the damping coefficient of particle vibration. The damping coefficient depends on the solid’s physical properties, e.g., viscosity, the mass of the vibrational particles, etc.

### 2.4. Propagation of Longitudinal Wave in a Viscous Isotropic Solid

Considering the shape and size of the particle and the mechanical network corresponding to the spring oscillator model of particle vibration, the mass of the particle and the friction force subjected to it are proportional to the density and viscosity of the solid, respectively. The compliance coefficient of the viscous solid is inversely proportional to the stubborn coefficient of the spring vibrator, as shown in Figure 1b. Also, $R_{m}=a_{1}\eta_{11}$, $m=a_{2}\rho$, $m=a_{2}\rho$, $k_{c}=1/C_{m}=a_{3}c_{11}$, respectively, and $a_{1}$, $a_{2}$, and $a_{3}$ are some defined proportionality factors.

Equations (3) and (4) reveal that the acoustic-impulse response and system function provide insightful information on the inherent physical properties of viscous solid media. These functions can help analyze the physical phenomena generated by acoustic waves propagating inside dense solids, e.g., the generation of intrinsic noise, acoustic attenuation, and dispersion.

As shown in Figure 2, we can use the mechanic network corresponding to the spring oscillator model shown in Figure 1b to describe the vibration state of a particle at any space position in viscous solid media and the propagation of the fluctuation. A vibrating particle located at a particular space point goes through internal stress, acting on the next particle near it and making it vibrate. This process is repeated, in turn, to achieve the propagation of acoustic waves inside a viscous solid medium. According to the mechanical network shown in Figure 1b, from Equations (21) and (23), we can obtain the vibration state of a particle at any space position inside viscous solid media by solving Equation (2).

The internal stress is now replacing the force (*f*_a_) in Equation (2) inside viscous solids, and the time variable is now *t*-*x*/*v_p_*. The internal stress induced by the vibration of the *j^th^* particle acts on the following particle (the (*j* + 1) *^th^* particle) to cause it to vibrate. The repetition of this process is the propagation of an acoustic wave.

## 3. Calculation and Analysis

In the following, according to the energy conservation law, we will compare, analyze, and discuss electromagnetic and acoustic waves in a few selected cases to understand the difference between electromagnetic and acoustic waves and the nature of the media.

**Case 1**: In ideal media

An ideal medium has zero electrical conductivity for the propagation of electromagnetic waves, and a perfect elastic medium has zero viscous coefficients for acoustic waves.

In an ideal medium, an electromagnetic wave source emits a continuous sinusoidal electromagnetic wave outward without propagation attenuation and damping attenuation.” The emitted electromagnetic signal enters space and propagates forward, i.e., the electromagnetic wave source provides electromagnetic energy propagating forward into space (following the energy conservation law).

In an ideal elastic medium, an acoustic source emits acoustic energy into the space, propagating forward without propagation attenuation and damping attenuation (following the energy conservation law).

**Case 2**: The power source emits a continuous sinusoidal wave in either non-ideal media (electrical conductivity is not zero) or non-ideal elastic media (viscosity coefficient is not zero).

For a non-ideal medium, the power source of the electromagnetic wave emits a continuous sinusoidal electromagnetic wave outward with propagation attenuation but no damping attenuation because an electromagnetic wave does not have mass. For a non-ideal elastic medium, the acoustic source emits successive sinusoidal acoustic waves outward, and there is only propagation attenuation without damping attenuation. The energy from an acoustic source includes the following three parts: (i) the energy lost from heat emission due to friction resistance during particle vibration, (ii) heat loss caused by wave propagation attenuation, i.e., heat loss by the viscosity of the propagation medium when the particle acts on the next particle through internal stress to make it vibrate, and (iii) the energy emitted to and contained in space, i.e., providing acoustic energy to propagate forward in space. Therefore, there is no damping attenuation of particle vibration in the time domain, only propagation attenuation of waves in the space domain, following the energy conservation law.

**Case 3.** The power source emits a signal wavelet in either non-ideal media (electrical conductivity is not zero) or non-ideal elastic media (viscosity coefficient is not zero).

The electromagnetic wave source emits an electromagnetic wave signal wavelet outward and propagates forward in the non-ideal medium with only propagation attenuation. The electromagnetic wave signal wavelet contains frequency components with different amplitudes, frequencies, and initial phases. Because the non-ideal medium is dispersive, the attenuation of an electromagnetic wave is a function of frequency in the propagation process. After some time, when the electromagnetic wavelet propagates to the next spatial position, propagation attenuation, reduced amplitude, and waveform distortion will occur. The energy of the transmitted electromagnetic wave signal wavelet has two parts; one part makes the electromagnetic wave signal wavelet continue to propagate in the non-ideal medium, and the other part produces heat loss, resulting in propagation attenuation.

For a non-ideal elastic medium, the acoustic source emits an acoustic signal wavelet outward with propagation attenuation and damping attenuation. When the acoustic wavelet signal propagates to a specific spatial location, it causes particle vibration at that location. Due to the viscosity of the medium and friction resistance, the particle vibration has damping attenuation in the time domain. The particle at a specific point in the space domain causes neighboring particle vibration through internal stress, yielding heat loss and propagation attenuation due to the viscosity of the medium.

The acoustic source emits a multifrequency acoustic signal wavelet. Due to the different propagation velocities and attenuation of various frequency components, the waves synthesized at other locations in space will have some degree of waveform distortion. Therefore, the propagation of the acoustic signal wavelet transmitted by the acoustic source in the dense medium has the damping attenuation of particle vibration and the propagation attenuation of the wave, which obeys the energy conservation law.

In summary, when an acoustic source emits a continuous sinusoidal acoustic wave outward, its waveform has only propagation attenuation. The constant emission of acoustic energy from the acoustic source makes up for the energy lost due to heat loss caused by medium viscosity in the process of particle vibration. In most practical systems, the emitted acoustic signal is a signal wavelet, i.e., the acoustic source radiates acoustic energy outward in a specific time interval. At the same time, it does not provide external power to supplement the energy loss caused by the particle vibrational damping decay (heat loss) in other time ranges. So, in the propagation process of an acoustic signal wavelet in a non-ideal medium, wave propagation attenuation and vibrational damping decay occur.

A gated-sinusoidal electromagnetic wave signal wavelet is shown in Figure 3a, propagating inside a non-ideal medium with only propagation attenuation for this electromagnetic wave signal wavelet. Figure 3b suggests that the gated-sinusoidal acoustic signal wavelet emitted by an acoustical source experiences the damping attenuation of particle vibration in the time domain and the propagation attenuation in the space domain inside a viscous solid. We cannot simply deduce the attenuation coefficient of an acoustic signal wavelet propagating in a viscous solid by the same method of driving the attenuation coefficient of an electromagnetic wave propagating in a non-ideal medium. For acoustic signal wavelets, we cannot simply neglect the effect of medium viscosity on the acoustic attenuation coefficient.

In the following, we selected Mesaverade sandstone (M-sandstone) as the solid sample to perform the calculation, comparison, analysis, and discussion. The density (ρ), related stiffness coefficient (*c*_11_), and viscosity coefficient ($\eta_{11}$) of M-sandstone are 2.710 kg/m^3^, 5.82 × 10^10^ N/m, and 8 × 10^4^
$N\cdot s/m^{2}$, respectively. The phase velocity of the longitudinal wave’s phase velocity ($v_{p}$) in a perfectly elastic rock is 4463 m/s, i.e., the case without considering the viscosity of M-sandstone.

### 3.1. Acoustic Impulse Response and Corresponding Amplitude Spectrum of Vibrating Particle

Based on Equations (3) and (4), using the physical parameters of M-sandstone and proportionality coefficients described above, the acoustic impulse response and corresponding amplitude spectrum of vibrating particles are calculated as shown in Figure 4, where the center frequency of the vibration particle in M-sandstone is 2.332 MHz.

Our calculated results also show that (i) the center frequency of a particle vibration system is only related to the stiffness coefficient of viscous solid media; (ii) the more significant the value of the stiffness coefficient, the higher the center frequency, which has nothing to do with its viscosity coefficient; (iii) the viscous coefficient only affects the duration of the acoustic impulse response; the more significant the viscosity is, the shorter the duration of the acoustic impulse response; the smaller the value of the amplitude spectrum.

Above all, at each spatial position, the acoustic impulse response and vibrational system function are related only to the physical properties of the solid and its internal structure, which have a specific effect on the attenuation coefficient of a longitudinal wave propagating inside a viscous solid medium.

### 3.2. Acoustic Attenuation Coefficient and Frequency Dispersion of Longitudinal Wave Propagating in a Viscous Solid

In this section, let us look at the acoustic attention coefficient and the frequency dispersion concerning the effect of medium viscosity. We will present the analysis and compare our calculations to the results from Auld [22] in Figure 5, Figure 6, Figure 7, Figure 8, Figure 9 and Figure 10.

For a longitudinal wave propagating inside a viscous solid medium, both the created attenuation and frequency dispersion are associated with the acoustic wave frequency and the solids’ physical parameters (elastic and viscosity coefficients). In other words, both of them are not only related to the propagation attenuation of an acoustic wave but also the damping attenuation of particle vibration.

By using the conventional method, such as Auld’s method [22], the derived attenuation coefficient for longitudinal waves (without considering the influence of damping attenuation of particle vibration on acoustic attenuation coefficient) is shown by the following:(27)αpa=ω22ρc1112η11c11

The corresponding phase coefficient and phase velocity are as follows:(28)kpa=ωρc11121+38ωη11c112−12
(29)$$v_{pa}=\frac{\omega }{k_{pa}}$$

According to the newly derived attenuation coefficient, phase coefficient, and phase velocity, i.e., Equations (21)–(23), the calculated relationships of $\alpha_{p}$, *k_p_*, and *v_p_* versus *f* for several different values of $\eta_{11}$ are shown in Figure 5a, Figure 7a, and Figure 9a, respectively. The plots of $\eta_{11}$ for several different values of *f* are shown in Figure 5b, Figure 7b, and Figure 9b, respectively.

In terms of the longitudinal wave’s acoustic attenuation coefficient, phase coefficient, and phase velocity derived from Auld’s method [22], i.e., Formulas (27)–(29), the calculated relationships of $\alpha_{pa}$, *k_pa_*, and *v_pa_* versus *f* for several different values of $\eta_{11}$ are shown in Figure 6a, Figure 8a, and Figure 10a. The relationships of $\alpha_{pa}$, *k_pa_*, and *v_pa_* verses $\eta_{11}$ for several values of *f* are presented in Figure 6b, Figure 8b, and Figure 10b.

Figure 5a and Figure 6a show that for a given value of $\eta_{11}$, both $\alpha_{p}$ and $\alpha_{pa}$ increase with the increased frequency (*f*). The increasing ratio of $\alpha_{pa}$ to *f* is more prominent or much larger than that of $\alpha_{p}$ to *f*. Figure 5b,c shows that, for a given *f*, $\alpha_{p}$ increases in the low-frequency range and decreases in the high-frequency range with $\eta_{11}$. Corresponding to a turning point of $\alpha_{p}$ increasing to decreasing, the higher the frequency (*f*) value is, the smaller the value $\eta_{11}$. We also observed that in the higher frequency range, the effect of *f* on $\alpha_{p}$ is much greater than that of $\eta_{11}$ on $\alpha_{p}$. Figure 6b shows that for a given *f*, $\alpha_{pa}$ increases with the value of $\eta_{11}$ monotonously, and this may be that the effect of particle vibrational damping attenuation on acoustic attenuation is not considered.

Figure 7a and Figure 8a show, for several values of $\eta_{11}$, that *k_p_* and *k_pa_* increase with *f*; the greater the value of $\eta_{11}$, the smaller the values of *k_p_* and *k_pa_*. Figure 7b and Figure 8b show that for several given values of *f*, (i) *k_p_* and *k_pa_* decrease when increasing $\eta_{11}$, and (ii) the greater the value of *f*, the greater the values of both *k_p_* and *k_pa_*.

In addition to the acoustic attenuation property of the longitudinal wave propagating in a viscous solid, we also pay attention to the dispersion phenomenon of phase velocity.

Figure 9a and Figure 10a show that for several given values of $\eta_{11}$, (i) both *v_p_* and *v_pa_* increase with *f*; (ii) the greater the value of $\eta_{11}$, the greater the dispersion degree of *v_p_*; (iii) the dispersion degree of *v_p_* is somewhat greater than that of *v_pa_*. Figure 9b and Figure 10b show that for several given values of *f*, (i) both *v_p_* and *v_pa_* increase with $\eta_{11}$, (ii) the higher the frequency, the greater the dispersion degree of *v_p_*, and (iii) the dispersion degree of *v_p_* is somewhat greater than that of *v_pa_*.

The above-calculated results show the differences between the various physical parameters $\alpha_{p}$ vs. $\alpha_{pa}$, *k_p_* vs. *k_pa_*, and *v_p_* vs. *v_pa_*. During the calculation of $\alpha_{p}$, *k_p_*, and *v_p_*, we consider both the effect of propagation attenuation and particle vibrational damping attenuation on them. In contrast, during the analyses of $\alpha_{pa}$, *k_pa_* and *v_pa_*, only the effect of the propagation attenuation of fluctuation is considered.

## 4. Conclusions

From the energy conservation perspective, we have analyzed the effect of particle vibration damping decay caused by viscosity on acoustic attention. Based on Auld’s theory, we introduced damping decay of particle vibration and derived the acoustic attenuation coefficient. Our new expression of the acoustic attenuation coefficient is an enhancement of Auld’s and the acoustic attenuation coefficient reported in the literature.

From theoretical derivation, calculation, discussion, and analysis, we conclude with the following remarks:(i)We analyzed the difference between acoustic and electromagnetic waves, i.e., the electromagnetic wave is a material with energy and without mass, and a vibration particle is a material with both mass and energy. Then, we derived new expressions of the acoustic attenuation coefficient ($\alpha_{p}$), phase coefficient (*k_p_*), and phase velocity (*v_p_*). These new expressions are different from conventional expressions. The reason for generating these differences may be that the newly derived acoustic attenuation coefficient ($\alpha_{p}$), phase coefficient (*k_p_*), and phase velocity (*v_p_*) consider the effects of both particle vibration-damping and fluctuation propagation attenuation on them; the conventional attenuation coefficient ($\alpha_{pa}$), phase coefficient (*k_pa_*), and phase velocity (*v_pa_*) only consider the effect of the fluctuation propagation attenuation on them.(ii)For all frequencies, the greater the value of $\eta_{11}$ is, the greater the attenuation coefficient ($\alpha_{p}$) and phase velocities (*v_pa_*, and *v_p_* ) and the smaller the phase coefficients (*k_p_* and *k_pa_*). In the higher frequency area, the larger the value of viscosity coefficient ($\eta_{11}$), the smaller the value of the attenuation coefficient ($\alpha_{p}$). In the lower frequency region, the larger the value of $\eta_{11}$, the larger the $\alpha_{p}$.(iii)For a given value of $\eta_{11}$, the values of $\alpha_{p}$, $\alpha_{pa}$, *v_p_*_,_ and *v_pa_* increase with frequency *f*, and the *k_p_* and *k_pa_* decrease with *f*;(iv)We also obtained the acoustic impulse response and vibrational system function corresponding to the longitudinal wave, which depends only on the viscous solid medium’s physical properties and internal structure.

In summary, the conventional expression of the acoustic attenuation coefficient is usually an analogy of the attenuation of an electromagnetic wave propagating in a non-ideal medium. It considers only the influence of fluctuation propagation attenuation on it. The new expression of the acoustic-attenuation coefficient reported in this article is related not only to the attenuation of the fluctuation propagation but also to the damping attenuation of particle vibration, which is different from that of the conventional acoustic attenuation. Because a vibrating particle inside a solid medium with viscosity has mass, energy, and inertia motion, it yields a short transition process and damping attenuation for particle vibration. The impulse response and system function can help invert the physical properties of the solid medium and the anomalies of the rock’s internal structure.

This report provided a more comprehensive explanation of the physical mechanism of the acoustic attenuation and frequency dispersion of longitudinal waves propagating inside viscous solids, which is essential to the forward study of inversion analysis of acoustical fields.

Based on Equations (21) and (23) and as an example of seismic exploration, we can modify the reflection coefficient of the elastic wave to improve the accuracy of amplitude versus offset (AVO) analysis. We can obtain a more accurate acoustic velocity of a longitudinal wave to perform the time-depth of seismic exploration data [25]. We can accurately convey the acoustic-velocity information of longitudinal waves to inverse the measured formation’s porosity in acoustic logging.

For the next step, exploring the attenuation pattern of the acoustic waves propagating inside anisotropic rocks should be interesting. As reported by Thomsen, some well-known rock anisotropic parameters are available for this purpose, e.g., [26]. Understanding the relationship between the transfer of interaction forces amongst particles and the propagation speed is also significant for dense solid media and solids with different porosities versus acoustic waves’ attenuation. These studies would enhance our understanding further for practical application.

## Figures and Tables

**Figure 1 micromachines-13-01526-f001:**
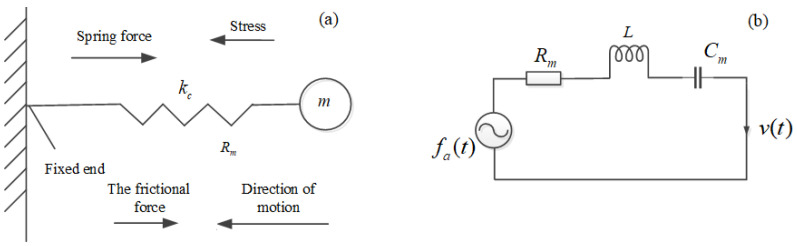
Mechanical analogy (spring oscillator) model of particle damping vibration corresponding to a longitudinal wave. (**a**) Longitudinal wave spring oscillator model; (**b**) Electromechanical analogical equivalent network.

**Figure 2 micromachines-13-01526-f002:**
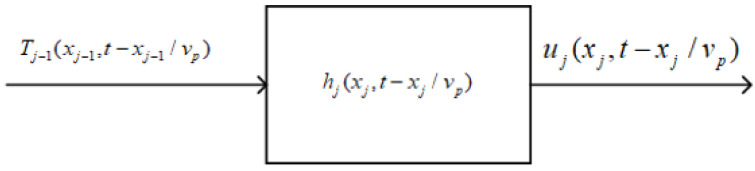
Propagation of an acoustic wave inside a viscous solid medium. $T_{j-1}(x_{j-1},t-x_{j-1}/v_{p})$ is the stress of the incoming vibrating particle acting on current particles; $h_{j}(x_{j},t-x_{j}/v_{p})$ is the impulse response corresponding to the particles inside a viscous solid medium; $u_{j}(x_{j},t-x_{j}/v_{p})$ is the particle displacement of the vibrating particle.

**Figure 3 micromachines-13-01526-f003:**
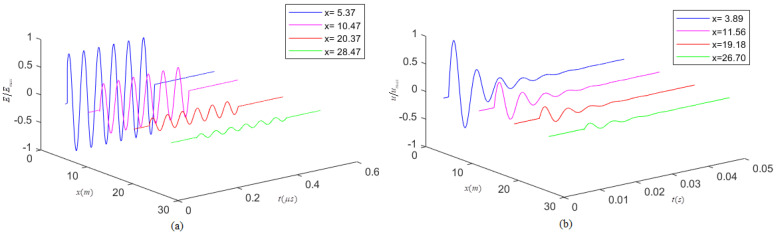
Schematic diagrams of an electromagnetic wave signal wavelet and acoustic signal wavelet propagating in the medium. *x* is the propagation distance of either electromagnetic wave signal wavelet or acoustic wavelet in the medium, and *t* is the propagation time. (**a**) An electromagnetic wave signal wavelet is inside a non-ideal medium. The vertical axis is the normalized amplitude of either electric field intensity or magnetic field intensity; (**b**) an acoustic wavelet propagating in viscous solids. The vertical axis is the normalized magnitude of the particle displacement.

**Figure 4 micromachines-13-01526-f004:**
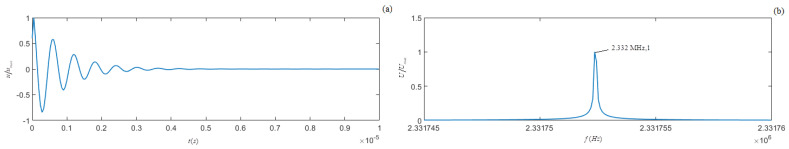
Acoustic impulse response and amplitude spectrum of vibration particle in M-sandstone. (**a**) The acoustic impulse response; (**b**) the amplitude spectrum.

**Figure 5 micromachines-13-01526-f005:**
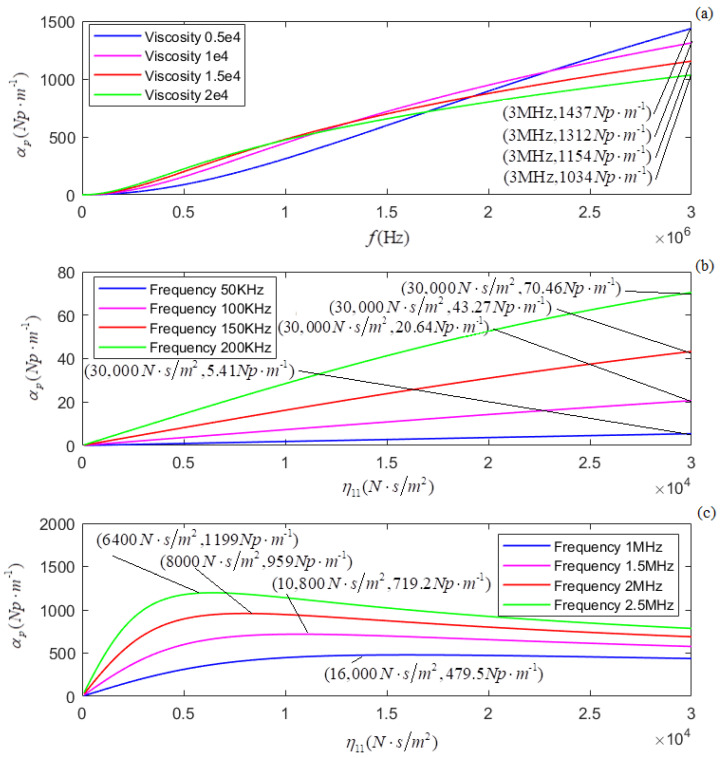
(**a**) The relationship between $\alpha_{p}$ and *f* for several different values of $\eta_{11}$; (**b**,**c**) the relationship between $\alpha_{p}$ and $\eta_{11}$ for several different values of *f*.

**Figure 6 micromachines-13-01526-f006:**
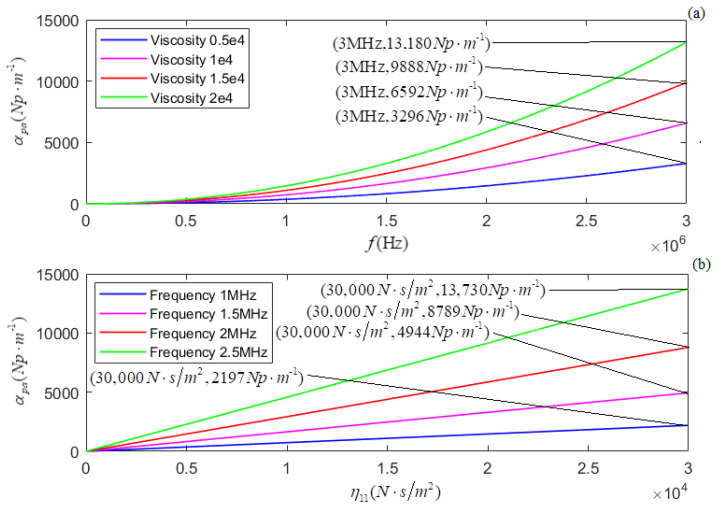
(**a**) The relationship between $\alpha_{pa}$ and *f* for several different values of $\eta_{11}$; (**b**) the relationship between $\alpha_{pa}$ and $\eta_{11}$ for several different values of *f*.

**Figure 7 micromachines-13-01526-f007:**
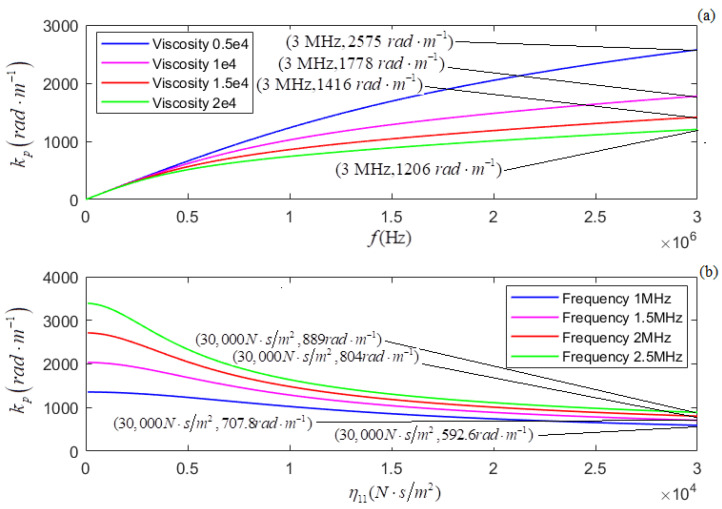
(**a**) The relationship between $k_{p}$ and *f* for several different values of $\eta_{11}$; ( (**b**) the relationship between $k_{p}$ and $\eta_{11}$ for several different values of *f*.

**Figure 8 micromachines-13-01526-f008:**
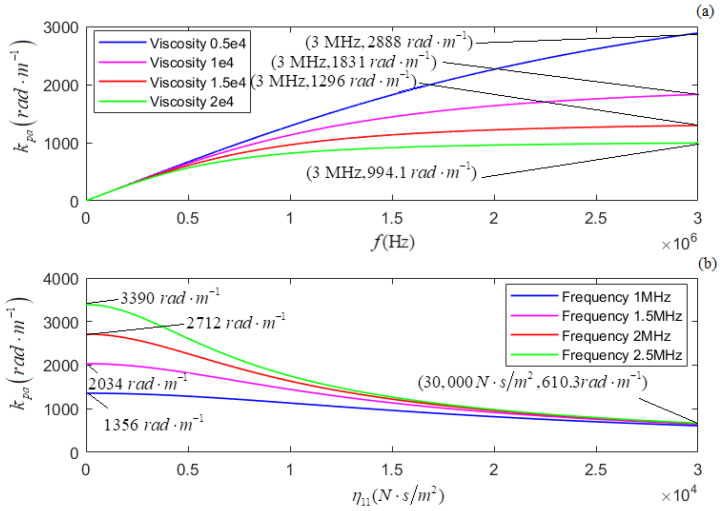
(**a**) The relationship of $k_{pa}$ versus *f* for several different values of $\eta_{11}$; (**b**) the relationship between $k_{pa}$ and $\eta_{11}$ for several different values of *f*.

**Figure 9 micromachines-13-01526-f009:**
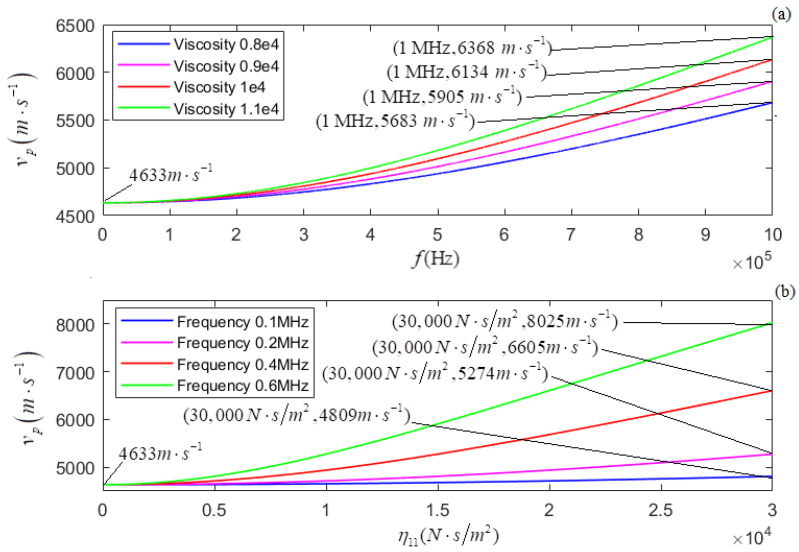
(**a**) The relationship of $v_{p}$ versus *f* for several different values of $\eta_{11}$; (**b**) the relationship between $v_{p}$ and $\eta_{11}$. for several different values of *f*.

**Figure 10 micromachines-13-01526-f010:**
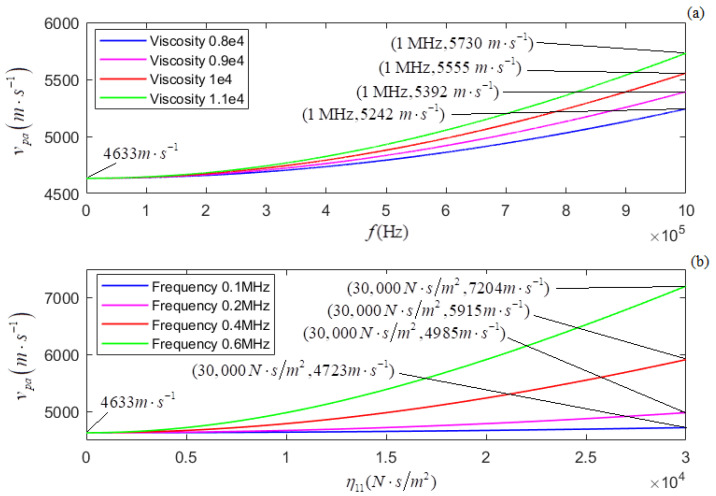
(**a**) The relationship of $v_{pa}$ versus *f* for several different values of $\eta_{11}$; (**b**) the relationship between $v_{pa}$ and $\eta_{11}$ for several different values of *f*.

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
