# Peer review of "Investigation of the Physical Mechanism of Acoustic Attenuation in Viscous Isotropic Solids"

_micromachines, 2022, doi:10.3390/mi13091526_

Round 1
Reviewer 1 Report
Manuscript ID:micromachines-1788250
Title:Investigation of the physical mechanism of acoustic attenuation in viscous solids
Authors:Lin Fa, Lili Li, Hong Gong, Wenhui Chen, Jing Jiang, GuoQiang You, Jifeng Liang, Yandong Zhang, MeiShan Zhao
In this work, the authors investigate the physical mechanism of acoustic attenuation of elastic waves propagating in viscous solids. An oscillator model with damping force proportional to particle velocity is used to explain the attenuation due to particle vibration in the medium. They consider a longitudinally propagating wave with damped amplitude at the entrance of the medium, and calculate the wave attenuation in space and time according to this setup. The authors then obtained a new attenuation coefficient that is different from the known one from B. A. Auld’s book. The authors also calculated wavenumbers and the phase velocities, which are also different from the known results. The authors compared the case of EM wave propagating in absorbing media with that of elastic wave propagating in a viscous solid, and claim that they are fundamentally different. The authors think that this difference is due to the non-zero mass of the vibrating particle in the second case.
Although I believe that the study of wave attenuation in viscous solids is important and has practical use, I do not recommend publishing this paper in its current form and content, as I find the authors misunderstood their own work. If the authors can revise this manuscript and correctly explain what their results mean, I would reconsider the possibility of a recommendation.
Now I explain why I think the authors misunderstood their own work.
1. The results in Eq.(21)-(23) are derived from the ansatz in Eq.(9). Note that at the entrance of the medium, that is, at x=0, it is assumed that the displacement u (x=0,t) has a decay factor exp(-beta*t), so the displacement u(x,t) at any position x>0 also carries this decay factor during time evolution. Without this decay factor, all the results below would reduce to the traditional known results as those in Auld’s book. However, I would say that this is just a choice of “boundary condition” or “initial condition”, and does not imply any essential difference between an absorbing medium for EM waves and a viscous solid for elastic waves. If we consider the displacement at the entrance as a “wave source” or “boundary condition”, then the choice here is simply to replace the traditionally used real frequency source of frequency omega by a source of complex frequency omega’=omega+i*beta. In fact, we can play the same game with EM waves and get the time damping effect of propagating waves even in a vacuum.
2. Regarding the mass issue, I would like to say that the Lorentz model of the dispersive-absorbing EM medium is also based on the same kind of oscillator model, so there is no essential difference in the consideration of particle mass.
3. If you consider a pulse propagating in an absorbing medium, whether the wave is an wave or an elastic wave, you will get a damping pulse. Also, if you consider a stationary source of real frequency, you will only get a spatially decaying wave without time damping effect, because the source is stationary. Furthermore, if you consider a complex frequency source, you get an damping wave or exponentially amplified wave, depending on the sign of the imaginary part of the frequency.

Author Response
The uploaded file is response to Reviewer 1.

Reviewer 2 Report
Comments and Suggestions for Author
The Manuscript ID micromachines-1788250 titled “Investigation of the physical mechanism of acoustic attenuation in viscous solids” belongs to the field of fundamental physics-acoustics subdivision acoustic attenuation characteristics.
Concerning the scientific content of the paper
1. In paragraph “1. Introduction” is missing two basic articles in the field:
DOI: 10.1017/S002211200000272X, https://doi.org/10.1016/0301-9322(92)90053-J, therefore the paragraph must be developed,
2. In Figure 1 isn’t given a detailed explanation for the electric analogy (fig. 1 b.),
3. In paragraph “2. Physics Model” the authors considered only the case of “viscous isotropic solids” and therefore the title of the article must be changed to “Investigation of the physical mechanism of acoustic attenuation in viscous isotropic solids”,
4. In paragraph “3. Calculation and Analysis,” the results obtained based on the considered model in paragraph “2. Physics Model” are compared only with the results published in references [31], [36](this reference cannot yet access), the aspect being insufficient and irrelevant for such a subject. The obtained results are not compared with at least 5-6 published results in the field.
5. In paragraph “4. Conclusion,” the novelty brought by the paper is not emphasized. Also, any future work in the mentioned field of acoustics attenuation characteristics is not mentioned.
About the presentation of the paper concerning the Manuscript Type MDPI journal template
1. The manuscript respects the Manuscript Type MDPI journal template.
For the aspects mentioned above, I strongly advise the major revision of the paper.
Author Response
The uploaded file is the response to the comments and suggestions from Reviewer 2.

Reviewer 3 Report
In this work, the authors derive the acoustic attenuation coefficient from combining the dynamical equation of solid in an acoustic field with conventional longitudinal wave propagation using spring oscillator model and a frequency dispersion relation of phase velocity for the longitudinal wave propagating inside viscous solid media. They employ a rock sample for analysis and calculate the system impulse response and vibrational function. They found that the acoustic impulse response and the system function of vibration particles depended on the physical property of the viscous solid media and their internal structure.
In this way, this work further contributes to the development of acoustic attenuation theory and therefore can be considered for publication, in principle.
However, despite said above, I see several issues in the manuscript that prevent me from recommending this work for publication in its present form.
1) The authors repeatedly refer to the “traditional acoustic attenuation coefficient” as a quantity derived “from an analogy of attenuation of an electromagnetic wave propagating inside a non-ideal medium” as to an oversimplified and insufficient approach: “…particles inside viscous solids have mass, vibrating energy, viscosity, and the inertia of motion, and they go through transient and damping attenuation processes. Accordingly, the conventional acoustic attention coefficient does not reflect the physical realities in many practical applications.”
That indeed would be true if there were no further developments of the theory of acoustic attenuation that take in account the above effects. However, this is not the case. There are many published works on the topic that describe the above effects in relation to acoustic attenuation. For example, the paper of T.L. Szabo “The time domain wave equations for lossy media…” in J. Acoust. Soc. Am. 96, 491 (1994).
Regarding viscosity, I find this statement confusing as acoustic attenuation, by definition, refers to viscosity: “… acoustic attenuation. It refers to the total energy loss of ultrasound as a function of depth. Attenuation is mainly due to thermoviscous effects, although additional relaxation phenomena play also a role…” [M. Mischi, ... M.A. Averkiou, in Comprehensive Biomedical Physics, 2014].
Therefore, I find the motivation of the authors (claiming that the works on the topic published so far did not address the effects related to, e.g., viscosity, vibrations of the particles, inertia) to be not completely accurate and even confusing in some instances. In my opinion, the authors should accurately describe the state-of-the-art and define the place of their own results among other related works, with a clear statement of the added value of their results.
Some (out of many) works on the topic:
1. Alok Kumar Verma et al., Study of Ultrasonic Attenuation and Thermal Conduction in Bimetallic Gold/Platinum Nanofluids, Johnson Matthey Technol. Rev., 2021, 65(4), 556–567
2. Richard L. Gibson, Jr. and M. Nail Toksoz, VISCOUS ATTENUATION OF ACOUSTIC WAVES IN SUSPENSIONS, J. Acoust. Soc. Am. 85(5), May 1989, p. 1925-1934
3. Frank Babicka and Andreas Richter, Sound attenuation by small spheroidal particles due to visco-inertial coupling, J. Acoust. Soc. Am. 119(3), March 2006, p. 1441-1448
4. J. He, J. Sapriel, and R. Azoulay, Acoustic attenuation and optical-absorption effects on light scattering by acoustic phonons in superlattices, Phys. Rev. B 40(2), 15 JULY 1989-I, 1121
5. Marcel Frehner, Holger Steeb and Stefan M. Schmalholz, Wave Velocity Dispersion and Attenuation in Media Exhibiting Internal Oscillations, Source: Wave Propagation in Materials for Modern Applications, Book edited by: Andrey Petrin, ISBN 978-953-7619-65-7, pp. 526, January 2010, INTECH
2) Next, I find the results section (“3. Calculations and analysis”) rather poor. Again, it uses the invalid comparison to the case on non-decaying EM waves in the beginning. It is followed by a series of similar figures (Figs 5 to 10) that are presented as an array, one by one, without an analysis of what is plotted in those figures. The figures look similar, and I am not sure the reader could guess from those figures of the important physics behind the multiple similar curves. The authors provide no discussion.
Once the authors claim that their attenuation coefficients are different from “conventional”, it would be appropriate to compare their findings to the conventional attenuation coefficients and explain the advantage of using their results. It is also common to consider the transition of the newly found results to the conventional coefficients to reveal the essential difference. This has not been done, and no discussion of the results is provided.
Correspondingly, the conclusions are weak and qualitative.
3) The manuscript is quite carelessly written. This refers to both the stye and grammar. Some examples to be improved (just on one page):
p. 2: “… and so did Peter et al.[2]” (poor style),
p. 2: “Their studied results showed…” (grammar),
p. 2: “Similar to an electromagnetic wave propagating in non-ideal media and an acoustic wave propagating in a viscous medium, there are some similarities but some differences.” (poor language),
p. 2: “These studies did not combined…” (grammar)…
In conclusion, in my opinion, this manuscript cannot be recommended for publication in its present form (although potentially it might contain interesting results if presented in a better way). It requires substantial revision taking in account the comments, listed above, before it can be considered for publication in “Micromachines” or in any other journal.
Author Response
We appreciate the Reviewer's comment and the criticism, which leads to the improvement of the manuscript.

Round 2
Reviewer 1 Report
A stationary source at the entrance of the medium does not have the decay factor exp(-beta*t). You must explain why you have to choose a source having this decay factor and why the traditional source without this decay factor is not allowed to use.
Author Response
Dear Editor and Reviewer,
We appreciate your comments and suggestions.
Our manuscript Micromachines-1788250 has been commented on in the second round of review.
#2 reviewer agrees to accept.
#1 reviewer's comments and suggestions are as follows:
“A stationary source at the entrance of the medium does not have the decay factor exp(-beta*t). You must explain why you have to choose a source having this decay factor and why the traditional source without this decay factor is not allowed to use.”
According to the law of conservation of energy, we explain the difference of attenuation produced by electromagnetic wave and acoustic wave from the physical mechanism (See Upload file).
Best Regards
Fa, Lin
Professor of Physics

Reviewer 2 Report
Comments and Suggestions for Authors
The Manuscript ID micromachines-1788250 titled “Investigation of the physical mechanism of acoustic attenuation in viscous isotropic solids” belongs to the field of fundamental physics-acoustics subdivision acoustic attenuation characteristics.
Concerning the scientific content of the paper
I thank the authors that they take into consideration the comments and suggestions of the first review report. All the recommendations and suggestions were satisfied.
Therefore, I recommend the acceptance of the article for publication.
Author Response
We would like to thank the reviewer for his/her valuable comments and suggestions. By reading the two papers recommended by the reviewer, we have a great help to the improvement of the manuscript.
Reviewer 3 Report
Following the previous referee reports, I would like to point out two major issues that, in my opinion, require attention and improvement before the paper can be published.
1) The motivation
In the abstract, the authors state:
“The traditional acoustic attenuation coefficient was derived from an analogy of attenuation of an electromagnetic wave propagating inside a non-ideal medium, featuring only the attenuation of wave propagation. Nonetheless, the particles inside viscous solids have mass, vibrating energy, viscosity, and the inertia of motion, and they go through transient and damping attenuation processes. Accordingly, the conventional acoustic attention coefficient does not reflect the physical realities in many practical applications.”
I find this statement misleading. It can be understood that until now, only the “conventional acoustic attenuation coefficient” was known, and the effects related to mass, vibrating energy, viscosity, and the inertia of particles were not studied. This is not the case, as already mentioned in the previous referee report. Examples of papers studying effects related to mass, vibrating energy, viscosity, and the inertia of particles were listed in the previous report. For example, Babick and Richter [J. Acoust. Soc. Am. 119, 3, 1441 (2006)] analyzed the effect of visco-inertial coupling on sound attenuation. Next, Gibson and Toksoz [J. Acoust. Soc. Am. 85(5), 1925-1934 (1989)] proposed and analyzed a model for attenuation of acoustic waves in suspensions that includes an energy loss due to viscous fluid flow around spherical particles. Further, the effect of thermal conductivity was discussed in several papers including: [A. K. Verma et al., Study of Ultrasonic Attenuation and Thermal Conduction in Bimetallic Gold/Platinum Nanofluids, Johnson Matthey Technol. Rev., 2021, 65(4), 556–567]. These are just few examples. Many more papers have been published studying the effects related to mass and inertia of particles, viscosity of the medium and vibrational energy.
Thus, the state-of-the-art in the field covers the effects related to mass, vibrating energy, viscosity, and the inertia of particles. Therefore, it would be misleading to claim that these effects were not studied so far and thus could be a motivation to study them.
The authors repeatedly use this motivation throughout their manuscript. Thus, in the revised manuscript they write:
“We cannot simply neglect the medium viscosity on the acoustic attenuation coefficient for acoustic signal wavelets.”
Of course. Neglecting those effects would be oversimplification, and it would be incorrect. But they were described already in many studies (some examples are listed above). The question then is, what is new in this work as compared to the state-of-the-art?
2) The interpretation of results
As already mentioned in the previous referee report, there is no reasonable analysis of the results shown in Figures 5 to 10. No physics behind the presented curves is discussed. The authors mainly comment on the curves that they are “increasing” or “decreasing”, but this can be seen be the reader from the plots.
What would be important for the reader, is to understand WHY do those curves behave like that? Why do they show increasing or decreasing (or non-monotonic as in Fig. 5(c)) behavior depending on the parameters? This is not explained…
As a reader, I do not see what I can learn from those multiple plots. Detailed discussions should be added in the manuscript.
Next (this was also mentioned in the previous report), when one includes additional effects in the model, it is common to present the new findings and compare them to the “old” results obtained without these new effects (e.g., the Auld method the authors refer to). Then it becomes clear, how these additional effects influence the results. For example, if one analyzes the effect of inertia, it would be natural first to show the result for the case of zero inertia, and then gradually increase inertia and see how this influences the result.
The authors showed an attempt to compare their findings to the known results (obtained from Auld model). For example, in Figs. 5 and 6, they present results obtained from Auld model (Fig. 5) and their original results (Fig. 6). However, they do not provide an interpretation of the revealed difference (e.g., in lines 428-429): “A good explanation is that in the higher frequency range, the effect of f on α_p is much greater than that of \eta on α_p.” This is not an “explanation”, this is what we can see in the plot: this does not explain WHY the effect of f on α_p is much greater than that of \eta on α_p.
In other words, it would be highly desirable to provide discussions explaining the physics behind the new results, in comparison with the known results. For example, why is \alpha_p nearly linear (or sublinear) function of f (Fig. 5) while \alpha_ap is a parabolic function of f? Where the maximum in the function \alpha_p=f(\eta_11) comes from (what is the physics behind this effect?), and why it is absent in \alpha_ap=f(\eta_11)?
Adding these discussions is essential which would considerably improve the presentation and potentially increase the impact of this paper.
Author Response
We appreciate the comments and suggestions from Reviewer, which are decent, thoughtful, and helpful in improving the manuscript.

Round 3
Reviewer 3 Report
In the new revised version of the manuscript, the authors mainly addressed the comments of the previous referee reports. I find their responses and changes in the manuscript satisfactory, and therefore I recommend the latest version of the manuscript for publication.
However, before the paper is finally accepted, still few minor corrections should be made in the text. I guess these are linguistic issues, but they change the meaning of the statements (made them wrong) made by the authors. These mistakes must be corrected.
1) In lines 77-78, the authors write:
“As is known, an electromagnetic wave is an invisible substance with only energy without mass…”
This statement is wrong as an electromagnetic wave is NOT a substance. Substance is something that has mass. An EM wave does not have mass.
Better to write something like this:
“As is known, an electromagnetic wave has only energy but does not have mass…”
2) Again, in lines 113-114, the same statement is repeated:
“An electromagnetic wave is a substance with energy but without mass.”
Please correct, remove the word “substance”. An EM wave is not a substance.
3) Next, lines 320-322:
“For a non-ideal medium, the power source of the electromagnetic wave emits a continuous sinusoidal electromagnetic wave outward with propagation attenuation but no damping attenuation because an electromagnetic wave has mass.”
This sentence states that “an electromagnetic wave has mass”. This is wrong. It does not. Please correct, e.g., as follows:
“For a non-ideal medium, the power source of the electromagnetic wave emits a continuous sinusoidal electromagnetic wave outward with propagation attenuation but no damping attenuation because an electromagnetic does not have mass.”
Author Response
We would like to thank you for your valuable suggestions and modifications to improve our manuscript. We have replaced the original three sentences in the manuscript with the sentences you revised (marked in red).
Thanks again.